# Cell Proliferation in the Piriform Cortex of Rats with Motor Cortex Ablation Treated with Growth Hormone and Rehabilitation

**DOI:** 10.3390/ijms22115440

**Published:** 2021-05-21

**Authors:** Margarita Heredia, Virginia Sánchez-Robledo, Inés Gómez, José María Criado, Antonio de la Fuente, Jesús Devesa, Pablo Devesa, Adelaida Sánchez Riolobos

**Affiliations:** 1Department of Physiology and Pharmacology, Institute of Neurosciences of Castilla and León (INCyL), University of Salamanca, Avenida Alfonso X El Sabio s/n, 37007 Salamanca, Spain; robledo@usal.es (V.S.-R.); inesgomez_@usal.es (I.G.); jmcriado@usal.es (J.M.C.); jfuente@usal.es (A.d.l.F.); asriolob@usal.es (A.S.R.); 2Scientific Direction, Medical Center Foltra, Travesía de Montouto 24, 15894 Teo, Spain; 3Coordination and Research, School Medicine, Universidad del Valle, Cochabamba 1408, Bolivia; pdevesap@gmail.com

**Keywords:** motor ablation, growth hormone, rehabilitation, cell proliferation, piriform cortex, brain injury

## Abstract

Traumatic brain injury represents one of the main health problems in developed countries. Growth hormone (GH) and rehabilitation have been claimed to significantly contribute to the recovery of lost motor function after acquired brain injury, but the mechanisms by which this occurs are not well understood. In this work, we have investigated cell proliferation in the piriform cortex (PC) of adult rats with ablation of the frontal motor cortex treated with GH and rehabilitation, in order to evaluate if this region of the brain, related to the sense of smell, could be involved in benefits of GH treatment. Male rats were either ablated the frontal motor cortex in the dominant hemisphere or sham-operated and treated with GH or vehicle at 35 days post-injury (dpi) for five days. At 36 dpi, all rats received daily injections of bromodeoxyuridine (BrdU) for four days. We assessed motor function through the paw-reaching-for-food task. GH treatment and rehabilitation at 35 dpi significantly improved the motor deficit caused by the injury and promoted an increase of cell proliferation in the PC ipsilateral to the injury, which could be involved in the improvement observed. Cortical ablation promoted a greater number of BrdU+ cells in the piriform cortex that was maintained long-term, which could be involved in the compensatory mechanisms of the brain after injury.

## 1. Introduction

In 2020, traumatic brain injury (TBI) has been declared the third leading cause of death globally [1]. Traffic accidents are the most common reason of TBI [2]. TBI has great effects on public health and economic consequences since many of the affected people die or have serious long-term consequences when they suffer it severely [3].

In TBI, there are damages caused directly by external forces (primary damage) and others caused by physiological imbalances that occur in the brain as a result of the former (secondary damage) [1,4,5]. Primary damage refers to mechanical damage by stretching, compression and/or tearing of neurons, glial cells, axons and blood vessels [6,7]. Secondary damage refers to some pathophysiological events that feedback on the initial damage [8]. After undergoing mechanical damage, cells undergo necrosis and break down. This leads to the release of the excitatory neurotransmitter glutamate, which will bind to ionotropic receptors on target neurons [9]. The result is pathological depolarization and Ca^+2^ entry into cells. In response to excitotoxicity due to increased intracellular Ca^+2^, there is an activation of microglia and resident astrocytes, and of leukocytes and macrophages released by the breakdown of the blood-brain barrier and vascular endothelium. In this way, a glial scar will appear around the damaged area to regulate extracellular glutamate levels and protect against initial trauma [5,10].

However, excessive inflammation can cause neurodegenerative effects due to the production of cytokines and reactive oxygen species (ROS). The imbalance between ROS and antioxidants activates enzymes that cause changes in the permeability of membranes, damaging them [11,12]. The rupture of blood vessels by primary mechanisms will decrease the arrival of O_2_ to the brain tissues [13], which negatively affects glial cells adjacent to the lesion. This can cause edema due to structural damage and osmotic imbalance, increasing brain volume and intracranial pressure, which in turn will lead to increased cell death [14]. After TBI, there are alterations in neural networks and neurotransmitter systems that affect brain activity not only in the perilesional area, but also in other parts of the brain [8,10], sometimes far from the site of injury. 

The damage caused by TBI can be reversible, although full recovery is exceedingly difficult in severe cases. Primary damage is often irreversible and unpredictable and the only ways to avoid it are preventive measures [8,15]. To avoid secondary damage, it is essential to know it well to find therapeutic targets. It is important to apply treatments that correct more than one secondary mechanism, since these are many and complex; and do it at the right time, since each one acts in a different way and at different times [16]. Some of the treatments applied so far have been: anti-inflammatories, caspase inhibitors, progesterone, erythropoietin, antioxidants, neurotrophic factors (NGF, BDNF) [16,17], growth factors, and growth inhibitor blockers [18]. In addition to chemical treatments, it is essential to apply physical rehabilitation treatments to lost motor skills [19]. Previous studies in our laboratory, using embryonic tissue transplants for the recovery of severe injuries of the motor cortex, highlighted the functional recovery of the motor deficit caused by the injury when the animals were forced to use the hand affected by the injury (rehabilitative therapy). In addition, cortical tissue transplants established connections with the host’s adjacent cortex [20,21].

In recent years, growth hormone (GH) has been seen to be an important repair agent for brain damage [22,23]. Its receptor, GHR, is widely distributed in the brain [24,25,26] and has been shown that its cerebral expression increases after a brain injury [27]. Circulating GH can cross the blood-brain-barrier (BBB), but it is also expressed in neural cells, and peripheral GH administration may cooperate with locally produced GH, increasing the proliferative response of neural progenitors after a damage [28,29]. In the brain, GH induces the expression of multiple genes [25], including its own receptor [30], and even insulin-like growth factor (IGF-I) [31]. In addition to its neural expression, the free fraction of plasma IGF-I also crosses the BBB to perform specific functions in the brain [32]. GH and IGF-I have a synergistic effect on neurotransmitter activity, astrocytic communication, glucose metabolism, neuronal dendritic remodeling, angiogenesis [33], and for adequate brain growth and maturation. Therefore, endogenous GH/IGF-I axis or exogenously administered GH or IGF-I act in neuroprotection, neurogenesis, angiogenesis, and synaptogenesis [22,34,35,36,37,38,39], as well as modulate neurotransmission through the synuclein (SYN) signal [36] and increases key mechanisms for a series of brain functions [28,37,38,39] and brain repair after an injury [34,36,37]. These reasons led to the use of GH or IGF-I in different models of brain injuries [34,35,36,40,41,42,43,44,45,46,47,48,49,50,51,52,53]. Regardless of the fact that TBI can produce GH-deficiency (GHD) [42], which contributes to the cognitive sequelae associated with TBI [43], and GH treatment in rats after TBI improves their cognitive, motor and behavioral activities [45,46,47]. After GH administration, a direct and cooperative action of exogenous and local GH is observed [44]. There is an increase in the synthesis of BDNF and TrkB [36], an increase in the expression of the GHR gene [30], and an increase in the synthesis of GH in the progenitor stem cells, which causes their proliferation, both on the ipsilateral side as well as on the contralateral side to the lesion [45]. All of these effects reduce the symptoms caused by TBI and provide cognitive improvements [9,46,47], motor improvements, beneficial behavioral and psychological effects (reduces anxiety, depression, sleep disorders) and improves quality of life [48,49]. These positive results can be seen in both the short and long term and are like those seen with GH treatment in ischemia and stroke [50,51].

Rehabilitation therapy improves cognitive abilities such as attention and motor skills in those affected by stroke [52], and if this treatment is combined with the administration of GH, its effect will be enhanced. Previous studies in our laboratory in animals with severe ablation of the motor cortex highlighted the functional recovery of the motor deficit caused by the injury after treatment with GH and rehabilitation, and demonstrated that this recovery is activity-dependent. That is to say, it occurs when GH treatment is combined with rehabilitative therapy [45,53,54]. In addition, we have shown that the functional recovery of the motor deficit depends on the moment of administration of the GH treatment after the cortical injury [54]. 

The piriform cortex (PC) is part of the olfactory cortex involved in the sense of smell; a fundamental sense that warns animals of the dangers or the possibility of satisfying appetites such as thirst, hunger or sex [55,56]. Smell also triggers some behaviors such as feeding or defense [57]. The individual learns and remembers, thanks to the previous olfactory experience, how he has to respond appropriately to these stimuli, so PC requires direct connections with regions of the brain that modulate behavior, such as the hypothalamus and, later in evolution, the hippocampus [58]. 

The PC belongs to the paleocortex, the phylogenetically oldest cerebral cortex [59], being especially prominent in the brain of rodents [60]. The PC receives the connections from the olfactory bulb directly, without the information having passed previously through the thalamus, as occurs with the rest of the senses [61]. Furthermore, PC differs from the neocortex in that it presents an important self-associative architecture, with many interconnections between its subregions [62].

Sensory information from the ipsilateral nostril reaches the piriform cortex of each hemisphere [59]. In the nostrils are chemical receptors that are excited and send impulses to the olfactory bulb, from where it projects directly to the olfactory cortex through the lateral olfactory tract. The PC is the area of the brain that receives the most afferent connections from the olfactory bulb. The PC establishes efferent connections with a large number of brain areas: olfactory structures (primary olfactory cortex, pedicle cortex, amygdala, olfactory tubercle), the limbic system (thalamus and hypothalamus), the entorhinal cortex [63,64], the prefrontal cortex [65] and with the superior colliculus [66]. It also establishes important ipsilateral and reciprocal connections with the orbitofrontal cortex, which is a higher integrative center that encodes olfactory information and determines the significance of these stimuli, modulating behavior in response to them [67]. The PC stores long-term memories through reciprocal associative connections that occur in it [59,62]. For this, the great capacity for synaptic plasticity that it possesses and its connections with the hippocampus are particularly important. In the case of PC synaptic plasticity, it is a high precision system that allows the location of the olfactory stimulus in space and time [63]. These same memory storage processes occur in the hippocampus and, in fact, PC communicates with this area of the limbic system [59].

After brain damage, alterations occur that affect brain activity, not only in the peri-lesional area but also in other parts of the brain, far from the site of the injury [68]. Following GH treatment, increased cell proliferation has been reported in the corpus callosum, the striatum, the parietal cortex and the piriform cortex, the magnitude of which varies depending on the day of GH administration after injury [69].

The objective of this study was to investigate cell proliferation in the piriform cortex and its possible beneficious role after GH treatment and long-term rehabilitation after injury. For this, in animals with ablation of the motor cortex treated with GH at 35 days post-injury (35 dpi), we analyzed cell proliferation in the PC and compared it with that of animals with cortical ablation treated with GH in the short term (5 h after the injury), as well as that occurring in control animals with intact motor cortex.

## 2. Results

### 2.1. GH Treatment Plus Rehabilitation at Long-Term Post-Lesion (at 35 Dpi) Significantly Improved the Motor Impairment Induced by Cortical Ablation

From the paw-reaching-for-food task, the mean percentages of successful responses obtained with respect to the total number of responses and the mean of the total number of responses (successful plus unsuccessful with both paws) are shown in Figure 1.

In the presurgical sessions, the performance in the paw-reaching-for-food task was similar in all rats. All animals displayed a stable strategy using a single forelimb to reach the pellets between the third and sixth session. When the percentage of attempts using the right or left paw was between 85% and 100%, the rat was classified as a handled right- or left- referent. It was considered that animals were well trained when the percentage of successful responses was ≥60% during three consecutive sessions. The mean of the values obtained in the last three sessions was taken as reference.

To distribute the animals in the different experimental groups, the average of the results obtained in the last two sessions of this phase was taken. Therefore, as Figure 1 shows, both the percentage of successful responses (A) and the total number of responses (B) were similar in all experimental groups (Figure 1A, PRE, F_2, 12_ = 0.025; Figure 1B, PRE, F_2,12_ = 3.018).

On day 7 post-injury, when the effectiveness of the motor cortex injury was tested, the ANOVA test showed that there were very significant differences in the mean percentage of successful responses between the groups (F_2, 12_ = 36.856, *p* ≤ 0.0001); successful responses decreased significantly in all animals with lesion (LGH35 and LV35 groups) compared to the sham-operated control group, CV35, (Bonferroni *post-hoc* test, *p* ≤ 0.0001), as shown in Figure 1A (POST). The low percentage of successful responses was similar in all injured animals. Some of these injured animals changed their preferred paw, while others continued to use the preferred paw, but in all cases the number of successful responses clearly decreased.

Cortical ablation was severe and homogenous in size and localization, and similar to that made in previous studies from our group [20,53]. Lesions were restricted to the primary and secondary motor cortex areas (M1 and M2), although in some cases the cingulate cortex, area 1, (Cg1), was slightly affected (see Figure 8 in Materials and Methods; a schematic representation of a cortical lesion by aspiration is shown in the reference [53]).

The first period of rehabilitative therapy began on day 8 post-injury, with daily sessions of 3 min duration, for nine days (Figure 1, Rehabilitative Therapy 1). The rehabilitative therapy consisted of the compulsory use of the preferred hand, determined in the presurgical phase, and started before treatment with GH or vehicle solution. In this first rehabilitation period none of the injured animals showed improvement (LGH35, LV35) (Figure 1A). The two-way ANOVA (group x session) showed significant differences between the groups (F_2_, _12_ = 34.019, *p* ≤ 0.0001). Furthermore, a significant session-effect was observed (F_8_, _12_ = 3.893, *p* ≤ 0.0005). However, the interaction between the group and the session was not significant (F_16_, _96_ = 0.543). Daily ANOVA tests showed significant differences between groups in all rehabilitation sessions. The sham-operated animals (CV35 group) maintained the percentage of correct responses at a level similar to that obtained preoperatively; while the two groups of injured animals, (LGH35 and LV35 groups), showed a significantly lower level of correct responses. The Bonferroni *post hoc* test revealed that all the animals with lesions (LGH35 and LV35 groups) did not show improvement in motor deficit, their percentages of correct responses were similar between the two groups and statistically reduced compared to sham-operated animals (CV35 group) during the 9 sessions (Bonferroni test: minimum level of significance between groups *p* < 0.0167) (Figure 1, Rehabilitative Therapy 1). At the end of this first period of rehabilitation, all animals were kept in their cages with food and water *ad libitum.*

The second period of rehabilitative therapy began 35 dpi, coinciding with the start of treatment with GH (LGH35 group) or with the vehicle solution (LV35 and CV35 groups). In this second rehabilitation period, a marked improvement in the deficit of manual skill of the animals treated with GH (LGH35) was highlighted, the percentage of correct responses being similar to that of the control animals (CV35) during the nine sessions, except in the first session. In contrast, vehicle-treated lesion animals (LV35) maintained a statistically reduced percentage of successful responses compared to control animals (CV35) during the nine sessions (Figure 1, Rehabilitative Therapy 2). At 36 dpi, all animals began with daily bromodeoxyuridine (BrdU) injections, for four consecutive days (see experimental design in Material and Methods). In this second period of rehabilitative therapy with the forced use of the impaired paw, ANOVA (group x session) showed that there were significant differences between groups in the percentage of successful responses (F_2_, _12_ = 7.942, *p* ≤ 0.006). A significant session-effect (F_8_, _12_ = 5.580, *p* ≤ 0.0001) and a significant group and session interaction (F_16_, _96_ = 3.818, *p* ≤ 0.0001) were observed. The daily ANOVAs showed significant differences between the groups in all sessions. The Bonferroni *post hoc* test showed that the percentage of successful responses of animals treated with GH was similar to that of the control animals from the second to the ninth session, without statistically significant differences. In contrast, lesion animals treated with vehicle (LV35) maintained a statistically reduced percentage of successful responses compared to control animals (CV35) during the nine sessions (Figure 1, Rehabilitative Therapy 2).

Regarding the total number of responses, the ANOVA test showed that there were no significant differences between the different experimental groups in all phases of the experiment (Figure 1B). 

### 2.2. Cortical Ablation Promoted Cell Proliferation in the Piriform Cortex

Cell proliferation in the PC was analyzed by counting cells positive for BrdU (BrdU+) in 6 animals. Three of them were animals belonging to the behavioral study, each one being behaviorally representative of each of the three groups of animals investigated (LGH35, LV35 and CV35). The other three animals, in which cell proliferation in the piriform cortex was investigated, belonged to previous behavioral studies in our laboratory [45]. That is to say, animals with cortical ablation treated with GH or vehicle solution immediately after injury (5 h post injury) and sham-operated control (LGH1, LV1 and CV1, respectively). This comparative study has allowed us to investigate whether the cell proliferation of the piriform cortex depends on the time of initiation of GH treatment.

First, we investigated whether cell proliferation in the PC was similar in both hemispheres. In this sense, in each of the animals (experimental and controls), it was studied whether there were differences in the number of BrdU+ cells in the PC ipsilateral to the lesion compared to the PC contralateral to the lesion (undamaged hemisphere). The results obtained showed the existence of cellular proliferation in the PC of the control animals with undamaged motor cortex (CV1 and CV35), although no differences were observed between both hemispheres (Figure 2A,B). In injured animals treated with GH or vehicle solution, the number of BrdU + cells in the ipsilateral PC was similar to that of the contralateral cortex (Figure 2C–E), except in the animal treated with GH at 35 dpi, in which a significant increase in the number of BrdU+ cells was observed in the PC ipsilateral to the lesion compared to the contralateral PC (Figure 2F). 

The one-way ANOVA of the number of proliferative BrdU+ cells corresponding to all the animal studied revealed a marked increase in BrdU+ cells in PC, both ipsilateral and contralateral to the lesion, of the rat treated with GH at 35 dpi (LGH35) compared to the rat that received the hormone 5 h after the lesion (LGH1) (Figure 3A,B). No significant differences were observed in the number of BrdU+ cells between sham control animals with different times of BrdU administration, on day 1 post-sham operation (CV1) or on day 36 after sham injury (CV35) in both hemispheres (Figure 3A,B). Likewise, in animals treated with vehicle (LV1 and LV35), the number of proliferating BrdU+ cells in the PC was similar in both hemispheres (Figure 3A,B). 

We analyzed whether there were differences in cell proliferation in PC between control animals with intact motor cortex and injured rats treated with GH or vehicle. The aim of this analysis was to evaluate the magnitude of the changes that both cortical ablation and GH treatment could have caused in the proliferation of PC cells. 

In the ipsilateral PC of the injured animal treated with vehicle 5 h after the injury (LV1), a significant increase in the number of BrdU+ cells was observed, compared to the control rat (CV1) and the rat treated with GH (LGH1) (Figure 4A). However, there were no significant differences between the rat treated with GH and the control animal (Figure 4A). In injured animals treated with vehicle or GH at 35 dpi, a significant increase in the number of BrdU+ cells was observed in the PC ipsilateral to the injury, compared to the control rat (CV35) (Figure 4B). Likewise, the rat treated with GH (LGH35) showed a significant increase in the number of BrdU+ cells compared to the rat treated with vehicle (LV35) (Figure 4B).

In the contralateral PC, the results obtained were similar to those observed in the ipsilateral PC. Cell proliferation in the injured animals treated with vehicle at 5 h after injury (LV1), showed a significant increase compared to control (CV1) and GH (LGH1) treated rat (Figure 5A). In animals treated with GH or vehicle at 35 dpi (LG35 and LV35), a significant increase in cell proliferation was observed in PC, compared to the control (CV35) (Figure 5B). However, unlike the results obtained in the ipsilateral cortex, no significant differences were observed in PC cell proliferation between the rat treated with GH (LGH35) and the rat treated with vehicle (LV35) (Figure 5B). 

Figure 6 shows microphotographs of coronal sections of the PC ipsilateral to the lesion, corresponding to all the experimental animals studied. In injured animals treated with GH or vehicle solution at 35 dpi (LGH35 and LV35, respectively) an increase in the number of BrdU+ cells was observed compared to the CV35 control, as reflected in the graph of the Figure 4B. In addition, it was evidenced that BrdU+ cells were located dispersed mainly in layers II and III of the PC. In the injured animal treated with vehicle 5 h after the injury (LV1), there was an increase in the number of BrdU+ cells compared to the animal treated with GH (LGH1) and the control (CV1), as shown in the graph of Figure 4A. 

## 3. Discussion

The present study describes important results on the benefits of GH treatment and long-term rehabilitation after ablation of the motor cortex; and on the effect of cortical ablation on the cell proliferation of the piriform cortex. In animals with ablation of the motor cortex, treatment with GH + rehabilitation at 35 dpi induced a significant improvement in the fine motor deficit caused by the injury. The ablation of the motor cortex promoted an increase in the cellular proliferation of the PC of both hemispheres, which was prolonged in the long term after the injury. GH treatment at 35 dpi, induced an increase in PC cell proliferation ipsilateral to the lesion. 

The improvement in motor deficit observed after treatment with GH + long-term rehabilitation after cortical ablation (at 35 dpi) confirms previous studies from our laboratory [54], which show that the benefits of GH treatment depend on the time of initiation of treatment after the injury. Thus, at 7 and 35 dpi, treatment with GH + rehabilitation can functionally recover/improve the manual skill deficit induced by the injury, while at 14 dpi there is no improvement [54]. Although the brain at 14 dpi is not susceptible to receiving the benefits of GH treatment, in the long term (at 35 dpi) it is again receptive to the beneficial effects of GH administration. These findings could suggest that the long-term post-injury brain has reached a new level of homeostasis that would make it receptive to the benefits of GH treatment again. The improvement observed with GH treatment at 35 dpi questions the old postulates that advocated that improvement/recovery after brain injury occurs in a window of time after injury, without, after this time, there may be more possibilities for improvements. On the contrary, the injured animals treated with vehicle at 35 dpi showed a trend towards a slight improvement that was highlighted by a decrease in significance with respect to the control group in the paw reaching task that would be induced by the rehabilitation therapy itself. However, this slight rehabilitation-induced improvement was markedly increased by the synergistic action of GH treatment. This provides new data to add to the many already described in the Introduction regarding the multiple beneficial actions of this hormone at the brain level. 

Our results demonstrate the existence of cellular proliferation in PC from control animals with undamaged motor cortex. Classically, in the adult brain, the two best known and most studied neurogenic niches are the subgranular region of the dentate gyrus of the hippocampus (SGZ) and the subventricular zone of the lateral ventricles (SVZ). However, cell proliferation has also been detected in the brain in other areas, such as the hypothalamus, the amygdala or the piriform cortex [70,71]. In the latter, doublecortin positive cells (DCX+) (a marker of neurogenesis) have been detected. However, some of the physiological characteristics of these DCX+ cells in PC are typical of mature neurons, something that differentiates them from the DCX+ cells resident in conventional neurogenic niches. This may indicate that the DCX labeling of some PC cells expresses rather neuronal plasticity [60,72] and that this occurs even after maturation. On the other hand, when the presence of BrdU (a cell proliferation marker) was investigated in the PC, a co-localization was found with both DCX and NeuN (postmitotic marker of mature neurons) [72,73]. If the cell proliferation detected in the PC responds to the neural stem cells resident in the PC under physiological conditions, this would turn the PC into a new neurogenic niche [74,75]; although it could also correspond to proliferative immature cells migrated from conventional niches (SVZ) that would be stored in the PC in a latent state until their maturation was induced by activity (olfactory stimulation) [76,77,78,79]. However, this cell proliferation in PC could also be due to neuronal plasticity mechanisms [60]. Immature neurons are known to have proliferative capacity, although once they have matured they lose it. On the other hand, neuron precursor stem cell markers have not been detected in PC, as occurs in conventional niches, although oligodendrocyte precursor markers have been detected, but in small amounts [60]. Other authors have also demonstrated the existence of proliferative cells in PC under physiological conditions, although their number is quite low [78,80]. These findings agree with our results in control animals with intact motor cortex, in which we observed cell proliferation in the PC of both hemispheres, although, likewise, the number of proliferating cells, BrdU+, was small. To elucidate whether these proliferative PC cells in control animals correspond to immature cells migrated from the SVZ or to neuronal plasticity mechanisms, further studies would be necessary.

It is known that cell proliferation in PC increases when there is a lesion in the olfactory system, i.e., cell death in PC occurs due to alteration of its connection with the olfactory bulb or the limbic system, or due to loss of smell [60,72]. Under these conditions, the number of pyramidal neurons decreases dramatically due to injury, while the number of DCX + cells increases [80]. In our study, without specific lesion of the olfactory system, the ablation of the frontal motor cortex induced an increase in proliferating cells, BrdU+, in PC, a region far from the initial cortical lesion. Thus, in the animal with cortical ablation treated with vehicle 5 h after injury (LV1) or in injured animals treated with GH or vehicle at 35 dpi (LV35 and LGH35), a significant increase in the number of proliferating cells was observed. These results could be explained by the fact that after injury, LGH35 and LV35 animals are similar for 34 days; since during this period, both groups are injured animals awaiting treatment on day 35 post-injury. Therefore, cortical ablation produced an increase in cell proliferation in PC that was maintained long-term after the injury. This increase in proliferating cells could be integrated into neural plasticity mechanisms that could be activated after ablation of the motor cortex and that would be maintained in the long term after the injury, to compensate for the deficit caused by the injury. 

Treatment with GH at 35 dpi produced an increase in cellular proliferation in PC ipsilateral to the injury and a significant improvement in the deficit of manual skill caused by the injury. In an animal model of cortical stroke, hippocampal plasticity, a region remote from that in which the initial cortical stroke occurred, has been reported after GH treatment, as well as an improvement in cognition and motor function after stroke [50,51]. Therefore, this increase in ipsilateral PC cell proliferation could be involved in the motor improvement that we observed after GH treatment. GH treatment 5 h post-injury also produced a marked improvement in motor deficit [45]; but, however, a concomitant increase in PC cell proliferation was not observed. These data suggest that the different time of treatment with GH would be the cause of these variations in PC cell proliferation; since in the injured animals treated with vehicle (LV1 and LV35), these differences were not observed. Therefore, we could speculate that different mechanisms of action could participate in the effects of GH treatment, depending on the time of administration of the treatment.

Previous studies in our laboratory, in animals with cortical ablation treated with GH or vehicle at 5 h post-injury (LGH1, LV1), show a notable increase in cell proliferation in the perilesional cortex, this increase being greater in animals treated with vehicle [45]. Likewise, in the motor cortex contralateral to the lesion, a notable increase in proliferating cells is observed in animals treated with vehicle (LV1) [45]. The analysis of cellular proliferation in the perilesional and contralateral cortex in animals treated with GH or vehicle 35 days after the injury showed a notable decrease in the number of proliferating cells compared to the animals treated with GH or vehicle 5 h after the injury (data not shown). Therefore, in these cortical areas, cortical ablation promoted an increase in cell proliferation that was not maintained long-term after injury.

Although in the present study PC proliferation data have been obtained from only 6 animals, each one representative of the different experimental groups and controls investigated, this does not make them less valid; given that in these animals, the findings found would have participated in their behavioral results; since in our experiments, as in patients with TBI, each individual is unique and the mechanisms implemented after injury and those involved in the benefits of GH treatment may be unique or shared with other individuals. Since in our study we did not measure IGF-I in plasma or cerebrospinal fluid, we cannot rule out that this hormone and others (GH-induced) have participated in the achievement of the results obtained.

In any case, this study demonstrates once again the usefulness of the administration of GH, together with rehabilitation, in brain injuries, namely TBI in this case. 

## 4. Materials and Methods

### 4.1. Animals

The brains of 15 adult male rats of the Wistar breed (Charles River Laboratories, Spain), weighing 200–220 g at the beginning of the experiments, were used. The animals were kept under controlled temperature conditions (18–20 °C) and with a natural light/dark cycle. After arrival, animals were allowed to acclimate to the animal facilities for several days, before starting the experiments. They were fed with a normal chow diet and water ad libitum, except when the paw reaching-for-food task was performed. During this time, the animals were moderately food restricted until their body weight was reduced to 86–88% of their initial ad libitum weight. All the experiments and procedures developed were approved by the Bioethics Committee of the University of Salamanca (BIO/SA64/14, date 13/11/2014), and were carried out in accordance with the guidelines on animal care of the Council of the European Communities (2010/63/EU) and the Spanish Regulations (RD 53/2013 and law 32/2007). Effort was made to minimize the suffering and number of animals used, according to the European guidelines. 

### 4.2. Experimental Design

A time diagram of the experimental design is represented in Figure 7. 

The experimental design consisted of 5 phases:Pre-surgical phase: Behavioral test of fine motor skills.Ablation of frontal motor cortex. Evaluation of the effectiveness of the lesion (fine motor skill test).Treatment with GH or vehicle and rehabilitative therapies (forced use of the preferred paw).BrdU administration.

Surgical procedures and sacrifice were carried out under deep anesthesia with Equithesin (Chloral hydrate/magnesium sulfate/pentobarbital sodium; 20 mg/kg, i.p). 

#### 4.2.1. Presurgical Phase: Behavioral Test of Fine Motor Skills

Eight days after the animals arrived, they were trained for the paw-reaching-for-food task, a specific motor test for fine motor skills that we used in our previous studies [53,54]. In this test, carried out in a special cage, as described in previous studies from our group [20,53,54], the animals are conditioned to perform high-precision motor movements of extension and flexion of the fingers of the forelimbs to obtain food. Before testing, the animals were housed individually and, as described above, feeding was restricted. In the paw-reaching-for-food test, rats were required to extend a forelimb through the hole, grasp and retrieve a pellet from the groove, put it their mouths, and eat it, as depicted in previous studies from our group [53,54]. 

Each time an animal succeeded to eat a pellet without dropping it, it was counted as a successful response [45,54]. Dropping the pellet after grasping or raking it was considered an unsuccessful response. The percentage of correct responses was quantified with respect to the total number of responses (correct + incorrect with both hands). In this phase, the preferred hand of each animal was observed (when the percentage of attempts using the right or left hand was between 85% and 100%), which made it possible to designate the animals as right-handed and left-handed. It was considered that animals were well trained when the percentage of successful responses was ≥60%. The mean of the values obtained in the last three sessions was taken as reference. 

To distribute the animals in the different experimental groups, the average of the results obtained in the last two sessions of this phase was taken. 

#### 4.2.2. Ablation of Frontal Motor Cortex. Evaluation of The Effectiveness of the Lesion

Frontal motor cortex ablation was performed as in other studies by our group [45,53,54]. Briefly, on day 0 the rats were deeply anesthetized with Equithesin and subjected to motor cortical ablation or sham-injury (cortical ablation, *n* = 10; sham operated, *n* = 5). The lesion was performed unilaterally by aspiration in the motor cortex contralateral to the preferential hand (established in the preoperative phase). The lesion was performed according to the stereotactic coordinates of the motor cortex corresponding to the area of the anterior extremity of the motor cortex: anteroposterior (AP) = 1–4 mm anterior to bregma; laterality (L) = 1–3.5 mm with respect to the midline. The ventral limit of the lesion was the corpus callosum. The typical size and location of cortical ablation resulting from this lesion has been described [53] and shown in Figure 8. 

The study of the histopathological characteristics of the lesion of the frontal motor cortex, by means of immunohistochemistry of glial fibrillar acid protein (GFAP) and ionized calcium-binding adapter molecule 1 (Iba1), showed a marked gliosis in the perilesional cortex with the presence of reactive astrocytes, GFAP+ (Figure 9A,B). Furthermore, reactive microglia/macrophage-like cells, type Iba1 +, were also observed in the cortical area adjacent to the lesion (Figure 9C,D).

The effectiveness of the lesion was verified on day 7 after cortical ablation using the fine motor skill test. Animals with effective injury began to use their non-preferred paw to reach for food or the percentage of successful responses with the preferred paw was significantly reduced with regard to previous values in the presurgical phase. All injured animals had effective injuries. 

#### 4.2.3. Treatment with GH or Vehicle and Rehabilitative Therapies (Forced Use of the Affected Paw)

In one group of injured animals, rhGH (0.15 mg/kg/day. Saizen, Merck, Spain) was administered subcutaneously for 5 consecutive days, commencing on day 35 after cortical ablation (LGH35) (Figure 7A). Other group of lesion animals (LV35, *n* = 5) was given the same amount of vehicle (phosphate buffer saline 0.1 M, pH 7.4, PBS), following the same experimental protocol (Figure 7A). The rehabilitation therapy consisted in the forced use of the preferred hand (hand affected by the motor cortical ablation). To do this, a removable bracelet was placed on the non-preferred hand of the animal, which forced it to reach the food with the preferred hand, that is the hand affected by the injury, in the fine motor skill test (Figure 10). The bracelet impeded the rat from reaching the food with the non-preferred hand but no other movements with the limb. The animals wore the bracelet only during the test and not continuously. 

All animals, including control rats, underwent two periods of rehabilitative therapy, each lasting 9 days, in daily sessions of 3 min: a first period of rehabilitation that began on day 8 until day 16 after injury, and a second period that began on day 35 post-injury until day 43 post-injury (Figure 7A). Between the two periods of rehabilitation the animals were kept in their cages with food and water ad libitum. 

#### 4.2.4. BrdU Administration 

To evaluate cell proliferation, we used 5-bromo-2′-deoxyuridine (BrdU). As in previous studies, BrdU was used together with 5-fluoro-2′-deoxyuridine (FdU), an inhibitor of thymidine synthesis that allows the use of smaller doses of BrdU [45]. BrdU was dissolved in water at a concentration of 16 mg/mL. All animals, including controls, received a daily injection of BrdU/FdU (30 mg/kg BrdU/3 mg/kg FdU, Sigma), administered intraperitoneally (ip), for 4 days. Injections commenced on day 36 after the injury. No toxic effects of BrdU were detected along the experimental protocol. 

### 4.3. BrdU Immunohistochemistry

At the end of the second period of rehabilitative therapy, at 45 dpi, the rats were sacrificed by ip injection of Equithesin and transcardially perfused with 2% dextran in phosphate buffer 0.1 M, pH 7.4 (PB) followed by a fixative solution (4% paraformaldehyde in PB). The brains were dissected, post-fixed in the same fixative and cryoprotected in 30% sucrose in PBS. The brains were cut out on a freezing microtome, and 40 µm thick coronal sections were taken throughout the brain. Free-floating brain sections were used for immunohistochemical studies.

The sections were first incubated in 2N HCl for 30 min at 37 °C for antigen recovery, and subjected to several rinses with 0.1 M borate buffer, pH 8.5 for 30 min. After several rinses with Tris-phosphate buffer saline, pH 7.4 (TPBS), the sections were incubated in a block solution. The block solution included 3% horse normal serum and 0.2% Triton X-100 in TPBS. The sections were then incubated in primary antibody solution (mouse anti-BrdU, Abcam, 1:100 in TPBS with 1.5% horse normal serum) for 48 h, at 4 °C. After incubation with the primary antibody, the sections were rinsed several times in TPBS and then incubated in 0.3% hydrogen peroxide in TPBS for 12 min to inactivate endogenous peroxidase activity. After several rinsings with TPBS, the sections were placed on the secondary antibody (biotinylated anti-mouse 1:200, IgG; Vector Laboratories) in TPBS. Sections were rinsed several times in TPBS and incubated at room temperature for 2 h in a horseradish peroxidase complex (ABC Elite kit, Vector Laboratories). Next, immunoreactivity was visualized using 3–3′ diaminobenzidine tetrahydrochloride (DAB). The specificity of the labeling was verified in the control sections incubated without the primary antibody, in which no specific labeling was observed. Sections were mounted onto slides, stained with cresyl violet and coverslipped with Entellan.

### 4.4. Quantitative Analysis

Six animals (1 LGH35, 1 LV35, 1 CV35, 1LGH1, 1LV1 and 1CV1) were tested at each time point after the injury. A mean of twelve, 40 µm-thick, coronal sections were selected per brain hemisphere through the piriform cortex of each animal (from Bregma +0.72 mm to Bregma −2.52 mm). A systematic stereological approach was used to count BrdU positive cells (BrdU+), in the piriform cortex with the 5× objective. Cell counts were performed using Image J software (National Institutes of Health, Bethesda, MD, USA) compiled and analyzed. The count was carried out by a blind observer. The data were represented in the number of cells positive for BrdU/mm^2^.

### 4.5. Statistical Analysis

The analysis of the behavioral data obtained was carried out using the StatView program. The group of animals treated with GH was compared with the group treated with vehicle and the sham-operated control group. We compared the total number of responses (successful + unsuccessful with both paws) and the percentage of successful responses with the preferred paw with respect to the total number of responses. The results of fine motor skills were analyzed using a two-way analysis of variance (ANOVA) (group and session). When the global ANOVA showed a significant difference between the groups (*p* ≤ 0.05), a partial ANOVA was performed comparing the different groups in each session. The Bonferroni *post hoc* test was used (*p* ≤ 0.0167 was the value considered as the limit for significance between groups) to compare the individual means. The analysis of the BrdU immunohistochemical data obtained was carried out using the Prism-GraphPad program. ANOVA and Student´s t-test were used to compare means in immunohistochemical studies.

## 5. Conclusions

The data obtained in this study indicate that long-term GH treatment after ablation of the motor cortex induced a significant improvement in the deficit in manual skill caused by the injury, and an increase in cell proliferation of the piriform cortex ipsilateral to the injury. The improvement observed after long-term treatment with GH after injury calls into question classic concepts about the absence of improvement beyond a critical period after injury. The ablation of the motor cortex induced an increase in the cellular proliferation in the piriform cortex of both hemispheres that was maintained in the long term after the injury, and that could be involved in the mechanisms of compensatory brain plasticity, responsible for the functional improvements after a cortical injury. 

Finally, our data reinforce the concept that GH is a key hormone in brain recovery after damage, although its administration must be accompanied by adequate rehabilitation.

## Figures and Tables

**Figure 1 ijms-22-05440-f001:**
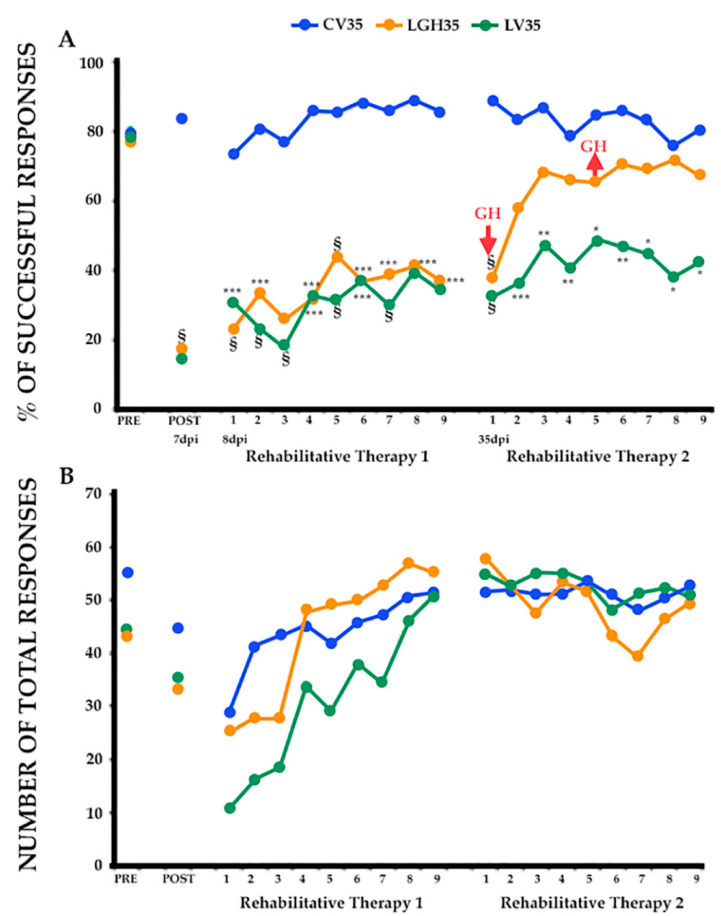
Effectiveness of GH treatment and rehabilitation 35 days after cortical ablation. Behavioral results obtained in the paw reaching-for-food task with the preferred paw (impaired paw) in the presurgical phase (PRE), post-ablation (POST) and rehabilitative therapies. (**A**). Mean percentage of successful responses (successful responses/total number of responses). (**B**). Mean of the total number of responses (successful and unsuccessful with both paws). The rehabilitative therapies consisted in the forced use of the impaired paw, in daily sessions for 3 min, for 9 consecutive days. GH-treated animals (LGH35) improved their percentage of successful responses after the second session (Rehabilitative Therapy 2), reaching a value no longer different from that of the sham-operated control group (CV35), while vehicle-treated animals (LV35) did not change its low percentage of successful results. Significance levels are obtained after comparison with sham-operated controls (CV35). § *p* < 0.0001; * *p* < 0.001; ** *p* < 0.005; *** *p* < 0.01 (Bonferroni test). GH arrows indicate when GH treatment commenced and finished in LGH35 animals. The x-axis indicates days.

**Figure 2 ijms-22-05440-f002:**
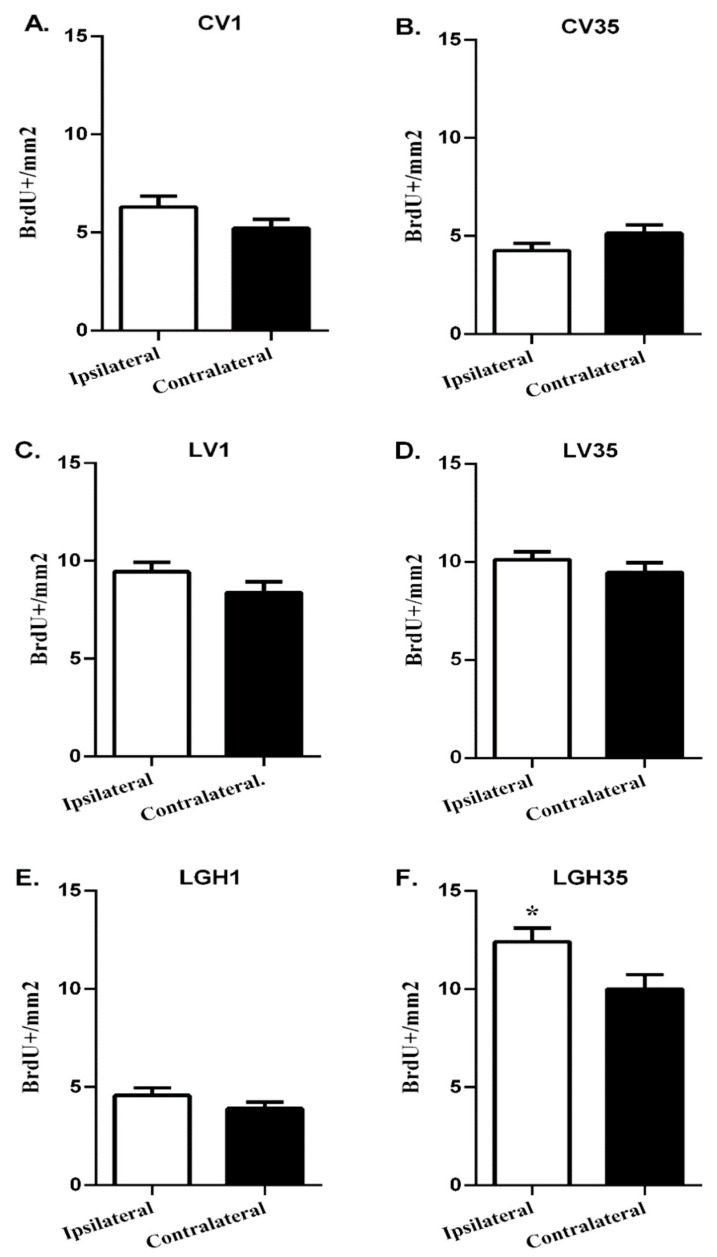
Cell proliferation in the piriform cortex ipsilateral and contralateral to the lesion, in the different animals studied. Comparison of the number of proliferative BrdU+ cells in the PC ipsilateral to the lesion with respect to the contralateral PC in each of the animals studied: (**A**) (CV1), (**B**) (CV35), (**C**) (LV1), (**D**) (LV35), (**E**) (LGH1) and (**F**) (LGH35). No significant differences were observed between both hemispheres in any of the animals, with the exception of the animal treated with GH 35 days after the injury (LGH35), which presented a significant increase in BrdU+ cells in the ipsilateral PC compared to the contralateral PC (**F**). Mean ± SEM; *n* = 12 images per hemisphere in each individual. * *p* < 0.05 (Student’s *t* test).

**Figure 3 ijms-22-05440-f003:**
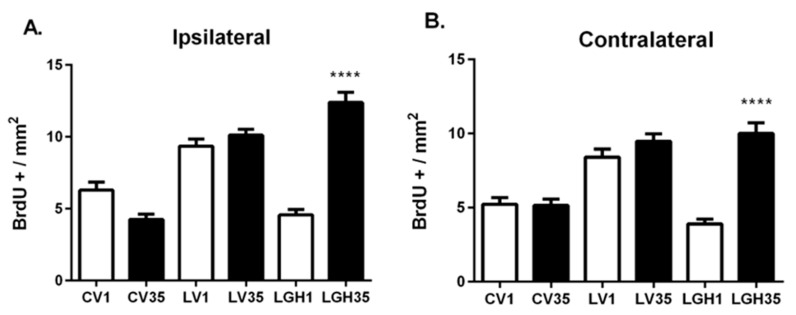
Cell proliferation in the ipsilateral and contralateral piriform cortex of controls and injured animals treated with GH or vehicle. One-way ANOVA revealed an increase in the proliferative BrdU+ cells in the animal treated with GH at 35 dpi (LGH35) both in the ipsilateral (**A**) and in the contralateral (**B**) hemisphere compared to the animal treated with GH 5 h after injury (LGH1). No significant differences were observed in the number of BrdU+ cells between the sham controls (CV1 vs. CV35), and between animals treated with vehicle (LV1 vs. LV35). Mean ± SEM; *n* = 12 images per hemisphere in each individual. **** *p* < 0.0001 (LGH35 vs. LGH1; ANOVA + Bonferroni *post hoc* test).

**Figure 4 ijms-22-05440-f004:**
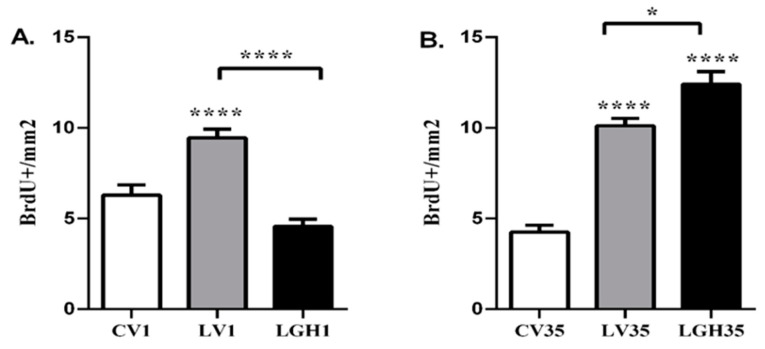
Cell proliferation in the ipsilateral PC of injured animals treated with GH or vehicle compared to the sham-operated control. (**A**). Treatment with vehicle or GH 5 h after the injury. Significant increase in the number of BrdU+ cells in the rat treated with vehicle (LV1) in relation to the control (CV1) and the rat treated with GH (LGH1). No significant differences were observed in the number of BrdU+ cells between the LGH1 rat and the control CV1. (**B**). Treatment with vehicle or GH at 35 dpi. Significant increase in the number of BrdU+ cells in the LV35 and LGH35 rats compared to the CV35 control. Likewise, a significant increase in BrdU+ cells was observed in the rat treated with GH (LGH35) compared to the rat treated with vehicle (LV35). Mean ± SEM; *n* = 12 images per hemisphere in each individual. * *p* < 0.05; **** *p* < 0.0001 (ANOVA + Bonferroni *post hoc* test).

**Figure 5 ijms-22-05440-f005:**
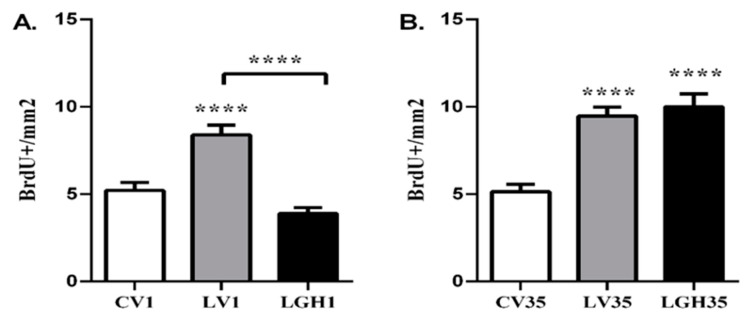
Cell proliferation in the contralateral PC of injured animals treated with vehicle or with GH compared to sham-operated control. (**A**). Treatment with vehicle or GH 5 h after the injury. In the rat treated with vehicle (LV1) a significant increase in BrdU+ cells was found in relation to the control (CV1) and the rat treated with GH (LGH1), while no differences were observed between the latter animals. (**B**). Treatment with vehicle or GH at 35 dpi. In both LV35 and GH35 rats a significant increase in the number of BrdU+ cells compared to the control (CV35) was observed. Mean ± SEM; *n* = 12 images per hemisphere in each individual; **** *p* < 0.0001 (ANOVA + Bonferroni *post hoc* test).

**Figure 6 ijms-22-05440-f006:**
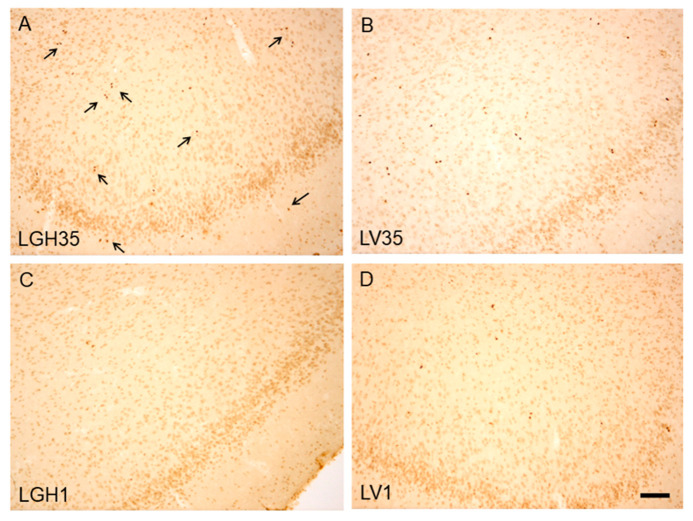
BrdU immunoreactivity in the piriform cortex ipsilateral to the lesion. (**A**,**B**). Injured animals treated with GH (LGH35) or vehicle (LV35) at 35 days after the lesion. (**C**,**D**) Injured animals treated with GH (LGH1) or vehicle (LV1) at 5 h post-injury. Black arrows in (**A)** indicate some proliferating BrdU+ cells. Coronal sections at AP = −1.08 mm relative to Bregma. Scale bar = 100 µm.

**Figure 7 ijms-22-05440-f007:**
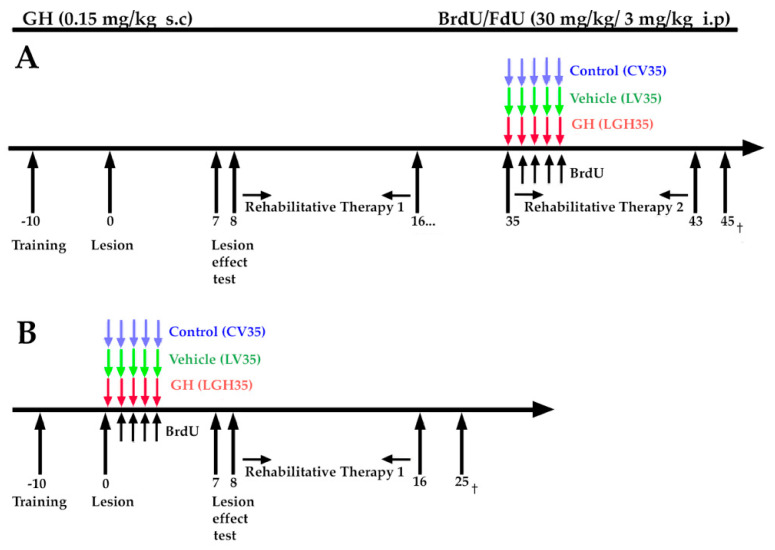
Time diagram of the experimental design. (**A**): Conditioning of the animals in the fine motor skill test for 10 days, prior to injury. Day 0: indicates the day of the injury or sham operation occurred (control animals). Evaluation of the effectiveness of the injury, through the fine motor skill test, on day 7 post-injury. Animals were treated with GH (0.15 mg/kg, s.c.) or vehicle at 35 days after injury (LG35, LV35 and CV35). BrdU/FdU injections (intraperitoneally) for 4 consecutive days, starting at day 36 after the injury. Application of two periods of rehabilitation therapy: one beginning on day 8 post-injury and the second (long-term) beginning on day 35 post-injury for 9 consecutive days. Slaughter of the animals on day 45 post-injury. (**B**)**:** Animals treated with GH or vehicle, or sham operated (LGH1, LV1, CV1, respectively), the same day of the injury (at 5 h post-injury), followed by rehabilitative therapy (days 8 to 16) and sacrifice 25 days after the lesion or sham-operation. Time scale: days. FdU: 5-fluoro-2′-deoxyuridine. †: sacrifice.

**Figure 8 ijms-22-05440-f008:**
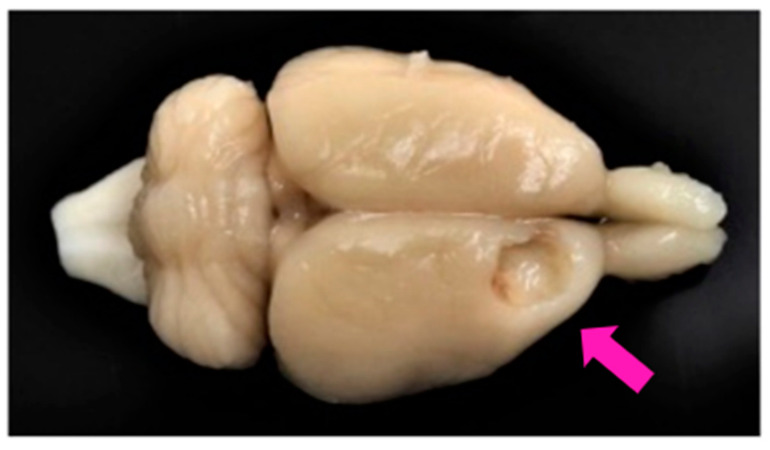
Frontal motor cortex ablation. Image of the brain of a left-handed rat with ablation in the right frontal motor cortex. At the bottom of the lesion, the corpus callosum can be seen, marking the ventral limit of suction. Arrow point out to the cortical ablation.

**Figure 9 ijms-22-05440-f009:**
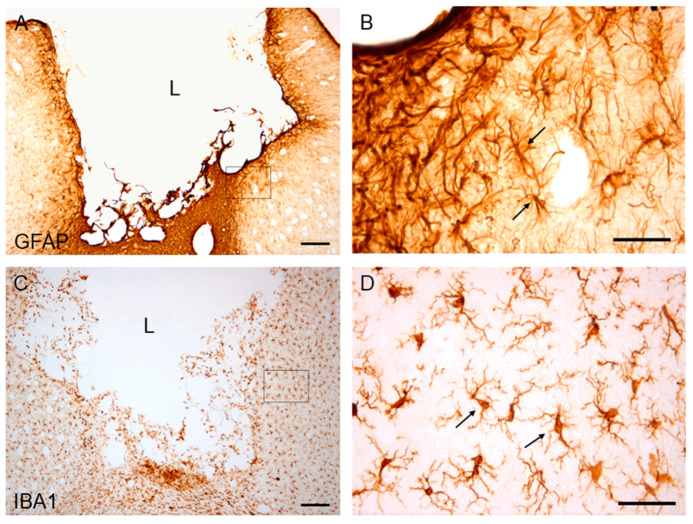
GFAP and Iba1 immunostaining in the perilesional cortical area of an animal with frontal motor cortex ablation. (**A**,**B**): GFAP immunoreactivity. (**C**,**D**): Iba1 immunostaininging. (**B**,**D**) are magnification of (**A**,**C**), respectively. The black arrows in (**B**) indicate some reactive astrocytes, GFAP+. The black arrows in (**D)** indicate reactive microglia/macrophage-like cells, Iba1+. Coronal sections at +2.04 mm from Bregma. Scale bars: 200 µm (**A**,**C**), 50 µm (**B**,**D**). (**L**): Lesion.

**Figure 10 ijms-22-05440-f010:**
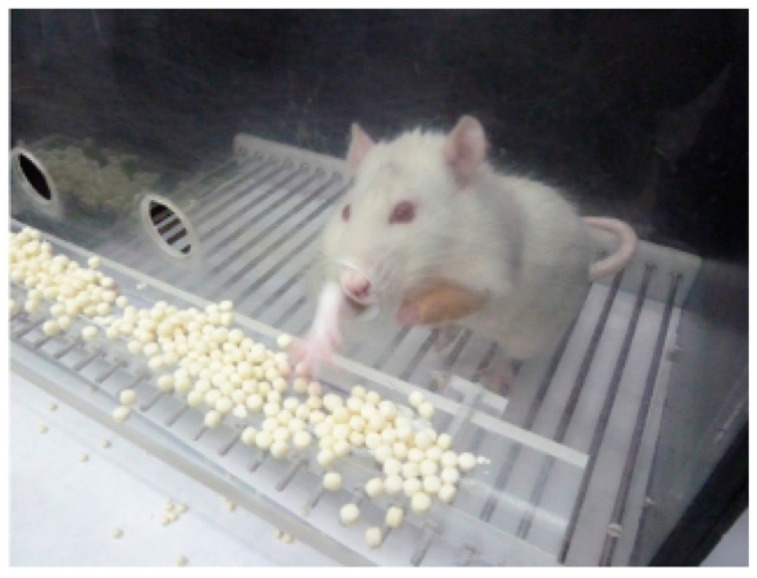
Rehabilitative therapy. The photograph shows a rat in the test cage with a plaster bracelet attached to the front paw of the non-preferred paw (intact paw, ipsilateral to the lesion), performing the paw-reaching-for-food task.

## Data Availability

Data can be found at the Department of Physiology, School of Medicine, University of Salamanca, Spain.

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
