# Peer review of "Cell Proliferation in the Piriform Cortex of Rats with Motor Cortex Ablation Treated with Growth Hormone and Rehabilitation"

_ijms, 2021, doi:10.3390/ijms22115440_

Round 1

Reviewer 1 Report

Congratulations to the authors for the manuscript: "Cell proliferation in the piriform cortex of rats with motor cortex ablation treated with growth hormone and rehabilitation". The data are clearly presented, the tables and figures well done. I suggest restricting the discussion on the basic concepts to be useful for the readers, particularly from the line 389 to 447 that report some concepts already explained in the results.

Author Response

Attached is the response.

Reviewer 2 Report

The study by Heredia et al assessed cell proliferation in the piriform cortex of post TBI treated with growth hormone and rehabilitation. the authors have assessed BRDU as a measure for proliferation, with the difference of having early BEDU injection or 35 post TBI.

the work has major concerns and misconception in the field of neurogenesis and rehabilitation along wrong experimental design.

First: the work lacks the  control group that lacks  what the authors consider "rehabilitation!!!; this is actually should not pas s the ICUC as you are inducing pain to the animal by preventing it from using the right arm !!!"

Second; early injection of BRDU and assessing at chronic time points measures proliferation and differentiation while  injecting at chronic time point assesses neurogenesis; I advice the authors to read about this

Third: the work has non conventional TBI model , nonconventional rehabilitation methods ....and These need justification and lack validation

Fourth: the work lacks cellular neural injury evaluation t show that this TBI model is valid; no behavioral assessment

Fifth, this injury in the PC region should be validated by the gain of olfaction after treatment.

basically, this work is based on  few images of the BRDU and making a paper out of it.

Author Response

Attached is the response.

Reviewer 3 Report

Minor comments: 

Please check for grammatical and spelling errors in the entire manuscript. 

Major Comments: 

  1. Can the authors combine figures 2, 3, 4 and 5 and perform ANOVA analysis? Using student's T test may not be the most appropriate statistical analysis. It is understood that the comparison is done between 2 groups, but essentially figures 3, 4 and 5 can be combined and ANOVA should be performed to check the actual effect of GH treatment.  
  2. The photomicrographs in Figure 6- The BrdU staining in panel A looks pretty insignificant. Can the authors provide better images that stand representative of the data presented? Given that the authors have performed a t-test to show a significance of p<0.05 (Figure 2F) and the effect is barely significant, the pictographs essentially also show that LGH35 has induced any significant proliferation of cells. 
  3. The authors either have to redo the statistics or find better images to represent the data. It would be much helpful if the statistics is redone. The images show that the effect is not significant. 

Author Response

Attached is the response.

Round 2

Reviewer 2 Report

we thank the authors for their reply,

as mentioned in my previous comments, a detailed histopathological analysis should be performed on the brain to show NeuN, GFAp, and iba1. the authors responded by repeating their writings and by references only a this is not acceptable.

BrdU staining should be repeated as it shows insignificance in the figures.

Author Response

Thank you for your comments. Please find the corrections we made according to your suggestions: 

as mentioned in my previous comments, a detailed histopathological analysis should be performed on the brain to show NeuN, GFAp, and iba1. the authors responded by repeating their writings and by references only a this is not acceptable.

As suggested by the reviewer, a histopathological study of the lesion of the motor cortex with antibodies against GFAP and Iba 1 has been carried out and a new figure has been made (Figure 9 in the revised manuscript). Likewise, in the revised manuscript on lines 581 to 585 a brief description of what was observed has been made.

BrdU staining should be repeated as it shows insignificance in the figures.

In the present work, each rat is unique since it has unique characteristics in its behavior throughout the entire experiment, from the validation of the effect of the injury to its recovery / improvement of the motor deficit, in some cases, or the absence of improvement. . in others, according to the treatment received. That is why repeating the BrdU studies would mean repeating all the work, which would take us almost a year of work. As suggested by the reviewer, a histopathological study of the motor cortex lesion with antibodies against GFAP and Iba 1 has been performed and a new figure has been made (Figure 9 in the revised manuscript). Likewise, in lines 581 to 585, a brief description of what was observed in the revised manuscript is made.

On the other hand, if the BrdU marking in the previous figure 6 appeared insignificant, it was due to the fact that this figure tried to represent the comparison between all the experimental and control rats, and therefore the photomicrographs had to be reduced in size to illustrate these difference in the same figure. To avoid this small problem, in this review a new figure 6 has been made comparing the marking with BrdU only in the experimental animals (LGH35, LV35, LGH1 and LV1), not presenting the photomicrographs corresponding to the control animals, which present very few proliferating cells BrdU +, as illustrated in other figures in the manuscript (Figures 2, 4 and 5). This has allowed us to increase the size of the photomicrographs and therefore make the BrdU marking more evident.

Reviewer 3 Report

Thank you for addressing all the concerns. 

Author Response

thank you for your suggestions that we corrected in the previous version. In this revised manuscript we included a new Figure 9 and corrected the Figure 6. 

Round 3

Reviewer 2 Report

thank you